# Flat band carrier confinement in magic-angle twisted bilayer graphene

Nikhil Tilak[1], Xinyuan Lai[1], Shuang Wu[1], Zhenyuan Zhang[1], Mingyu Xu[2,3], Raquel de Almeida Ribeiro [2,3], Paul C. Canfield[2,3] & Eva Y. Andrei [1✉]

Magic-angle twisted bilayer graphene has emerged as a powerful platform for studying strongly correlated electron physics, owing to its almost dispersionless low-energy bands and the ability to tune the band filling by electrostatic gating. Techniques to control the twist angle between graphene layers have led to rapid experimental progress but improving sample quality is essential for separating the delicate correlated electron physics from disorder effects. Owing to the 2D nature of the system and the relatively low carrier density, the samples are highly susceptible to small doping inhomogeneity which can drastically modify the local potential landscape. This potential disorder is distinct from the twist angle variation which has been studied elsewhere. Here, by using low temperature scanning tunneling spectroscopy and planar tunneling junction measurements, we demonstrate that flat bands in twisted bilayer graphene can amplify small doping inhomogeneity that surprisingly leads to carrier confinement, which in graphene could previously only be realized in the presence of a strong magnetic field.

[1] Department of Physics and Astronomy, Rutgers, The State University of New Jersey, Piscataway, NJ, USA. [2] Ames Laboratory, U.S. Department of Energy, Ames, IA, USA. [3] Department of Physics and Astronomy, Iowa State University, Ames, IA, USA. ✉email: eandrei@physics.rutgers.edu

When two sheets of graphene are superposed with a relative twist angle, they develop a moiré super-structure with an angle-dependent wavelength which reflects the local stacking variation of the two crystal lattices. Hybridization between the two sets of electronic bands produces a strongly modified, twist angle-dependent band structure[1,2]. Close to a "magic" angle (~1°), this leads to very narrow, almost dispersionless (flat) low-energy bands[3,4]. These flat bands are isolated from the dispersive bands by single-particle bandgaps[5]. Since the kinetic energy of the electrons in the flat bands is quenched, e–e interactions become important and give rise to phenomena such as correlated insulating states[6], superconductivity[7], emergent ferromagnetism[8,9], etc. The electronic properties of twisted bilayer graphene (TBG) were initially explored[1] in naturally occurring TBG synthesized by chemical vapor deposition, through the use of scanning tunneling microscopy/spectroscopy (STM/STS). More recently, the development of techniques[10] to control the twist angle and the doping level has expanded the range of capabilities to include global measurements such as magnetotransport[6,7,11–14], electronic compressibility[15], angle-resolved photoemission spectroscopy[16], as well as gated local probes including STM/STS[1,17–20], nano-SQUID-on-tip microscopy[21], local compressibility[22].

Obstacles to progress in this field include twist angle and doping inhomogeneity. Whereas efforts to address the former are being undertaken, the effects of local doping variation have thus far been ignored. Naively, this may be justified by the fact that spatial doping variations[7,11,21] in samples using hexagonal boron nitride (hBN) substrates[23] can be as low as $10^{10}$ cm$^{-2}$, which is two orders of magnitude lower than the typical charge density in magic-angle TBG, ~$10^{12}$ cm$^{-2}$. However, as we demonstrate below, even such low levels of density inhomogeneity can radically change the response and electronic properties of the system, obscuring the moiré physics when the Fermi level is aligned to the edge of the flat band.

Using tunneling experiments with a traditional STM as well as a novel planar tunneling device, we find that near the edges of the flat bands, the local doping variations which are ubiquitous in TBG devices, produce patches of conducting regions separated by insulating regions. This leads to carrier confinement on a scale typically larger than the moiré wavelength which can conceal the magic-angle physics.

## Results

We begin by discussing the results of tunneling measurements on a TBG device fabricated by a tear-and-stack technique[17] (see "Methods" for details). The schematic experimental setup is shown in Fig. 1a. We navigate to the micron size sample using a capacitance-based technique[24]. We identify a magic-angle region via STM topography and spectroscopy measurements. The honeycomb lattice of the carbon atoms in graphene is composed of two triangular sublattices labeled A and B. The flat bands are localized mostly on the AA stacking regions, where every atom in the top layer is positioned directly on top of an atom from the bottom layer. These appear as circular bright spots in the topography image when the flat bands are occupied[1] (Fig. 1b). Surrounding the AA regions are six darker regions called AB/BA. In the AB (Bernal stacked) regions, top layer A sublattice atoms are positioned directly above bottom B sublattice atoms while the top layer B sublattice atoms have no partners in the bottom layer. The BA regions are defined similarly via sublattice symmetry. These stacking arrangements are illustrated in Fig. 1c.

The local twist angle ($\theta$) is determined by measuring the average moiré wavelength ($L_M$) in the three crystallographic direction using the relation

$$L_M = a/((2\sin(\theta/2)), \tag{1}$$

where $a = 0.246$ nm is the lattice constant of monolayer graphene[3]. The data presented below were collected in a region with a twist angle of 1.12°, over at least 30 moiré unit cells. The heterostrain in the region was estimated to be 0.2% following reference[18].

Each flat band in TBG is fourfold degenerate owing to the valley and spin degrees of freedom. In total, it takes a carrier density of eight electrons per moiré cell to completely fill both the flat bands. For a twist angle $\theta$ this corresponds to a carrier density

$$2n_S \approx \frac{16}{a^2\sqrt{3}}\theta^2 \tag{2}$$

where $\theta$ is measured in radians. We can electrostatically control the carrier density by tuning the voltage ($V_g$) applied between the silicon backgate and the sample bias ($V_b$). The carrier density depends on $V_g$ as,

$$n = \frac{1}{e}\left(V_g - V_{g0}\right)\left(\frac{d_1}{\epsilon_0\epsilon_1} + \frac{d_2}{\epsilon_0\epsilon_2}\right)^{-1}. \tag{3}$$

Here $V_{g0}$ is the gate voltage needed to tune the system to charge neutrality, $\epsilon_0$ is the permittivity of vacuum, $e$ is the electronic charge, $d_i$ and $\epsilon_i$ are the thickness and dielectric constant where $i = 1, 2$ correspond to hBN and SiO$_2$, respectively. Given the geometry of our device and the measured twist angle, it takes 75–85 V of backgate to completely fill the empty flat bands.

Figure 1d shows a typical STS curve, which provides a measure of the local density of state (LDOS) within the AA regions when the flat bands are completely filled. The two sharp peaks in the LDOS correspond to two Van Hove singularities (VHS) in the electronic spectrum where the density of states diverges. The rapid decrease in the LDOS near the band edges suggests that close to the empty or full band the electronic properties are particularly susceptible to local doping variations, caused by twist angle inhomogeneity, impurities or defects. We measure the full width at half maximum of the electron and hole side flat bands to be ~16 meV and the two VHS are separated by ~18 meV. The dips in the $dI/dV_b$ between the flat bands and the remote bands correspond to the single-particle superlattice gaps[5].

Next, we measured the gate voltage dependence of the STS at an AA site (Fig. 2a). In addition to the flat band and the dispersive bands, a series of sharp peaks were observed in the $dI/dV_b$ spectra when the Fermi level was tuned close to the full filling of the electron-side flat band (Fig. 2c). These peaks, which are almost equidistant ($\Delta V_b \approx 60$ meV) in bias voltage, move toward higher bias values as the flat bands are filled. This is opposite to the expected gating behavior for normal features in the DOS which should evolve toward smaller bias values as the bands are filled.

Furthermore, when the peaks intersect the flat band at the Fermi level ($V_b = 0$ mV), a series of diamond-like structures appear in the gate dependence map which resembles Coulomb diamonds seen in quantum dots[25]. These can be seen more clearly in the zoomed-in view of the gate dependence map as shown in Fig. 2b. After eliminating trivial explanations of the origin of these peaks (such as dirt stuck to the tip), we concluded that the effect is intrinsic to the sample and not a tip artifact (see Supplementary Note 8). Recalling that, owing to the chiral nature of the quasiparticles, backscattering in graphene is suppressed, the observation of Coulomb diamonds may be surprising. In fact, Coulomb diamonds in graphene are not observed without applying a strong out-of-plane magnetic field. The magnetic field splits the bands into flat Landau levels separated from each other by an energy gap which depends on the magnitude of the applied magnetic field[26,27]. Split gates[28] or tip-induced band bending[29,30] can be used to locally bend the Landau levels which produce

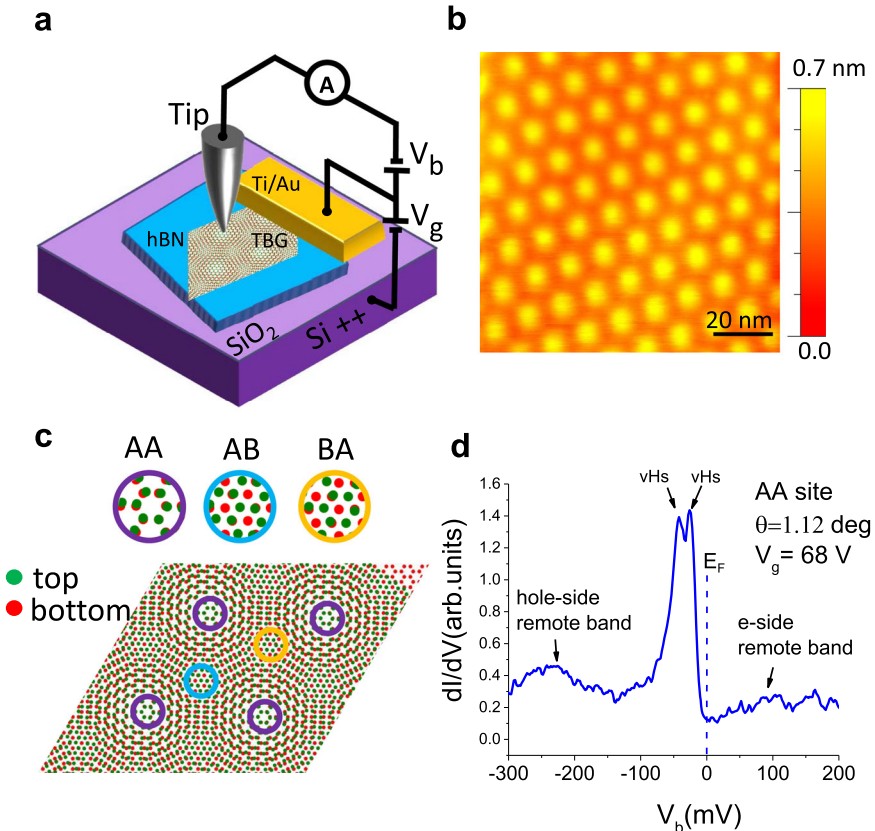

**Fig. 1 STM/ STS on magic-angle twisted bilayer graphene. a** Schematic of the STM device and measurement setup. The TBG sample was made using the "twist and stack" method. The Tungsten tip is kept grounded while a bias voltage $V_b$ is applied to the sample. A backgate voltage $V_g$ is applied between the p-doped silicon backgate and the sample. All measurements were taken at 4.9 K. **b** STM topography measured at a bias of −300 mV and tunneling current of −30 pA. The bright yellow spots which form a triangular lattice are the AA stacking regions. The surrounding darker orange regions have approximately AB and BA stacking. The moiré wavelength was measured from STM topography to be 12.6 nm which corresponds to a twist angle of 1.12°. **c** An illustration of the various stacking arrangements. The green and red dots represent the carbon atoms in the top and bottom Graphene layer respectively. **d** shows a typical $dI/dV_b$ spectrum measured at the center of an AA region at $V_g = 68$ V when the flat bands are completely full. The van Hove singularities (VHS) and the remote bands on either side of the flat band are labeled. Gap-like dips in the $dI/dV_b$ signal separate the flat bands from the remote bands. The Fermi level $E_F$ of sample is shown with a dashed line at $V_b = 0$ mV.

quantum dots in such samples, which are observed as Coulomb diamonds[31] in conductance vs gate voltage maps.

We show that similar to the Landau levels in graphene, the flat bands in magic-angle TBG amplify local density variations, but without the need of applying a magnetic field. To this end, we determined the gate voltage at which the flat bands are almost completely full, so that the Fermi level lies at the edge of the flat bands. At this gate voltage, 68 V, we collected spatial $dI/dV_b$ maps, shown in Fig. 3a for bias voltage $V_b = 0$ mV. These maps reveal prominent contrasting patches of bright and dark regions, corresponding to charge puddles created by the local doping inhomogeneity. A comparison of spectra (Fig. 3b) gathered in the bright and dark regions at positions marked by the green and red crosses respectively, shows the flat band is completely full in the dark regions while it is partially empty in the bright regions, directly confirming the spatial doping variation in this sample.

To further characterize this doping variation, we plot in Fig. 3c a line-cut across the conducting region in Fig. 3a along a path denoted by the yellow arrow. The energy of the charge neutrality point ($E_{CNP}$) is marked by the solid black line as a guide to the eye. In the absence of any doping variation, one would expect the $E_{CNP}$ to be constant regardless of the position for a given filling. The black line traces the shape of the potential well created by the doping disorder at this gate voltage. Since there are no electronic states available near $E_F$ in the regions immediately surrounding

the bright islands, the carriers in the bright regions should be confined within them. They can only occupy discrete energy levels whose separation depends on the shape and size of the potential well. Carriers from the tip can tunnel into the bright regions only into these discrete energy levels. These confined carriers can tunnel from the bright conducting islands to the nearby islands through the short insulating barrier separating them. We illustrate this situation with a tunneling diagram in Fig. 3d. Hereafter we refer to these conducting regions as quantum dots.

Valuable information about quantum dots can be extracted by analyzing their charging characteristics. Upon analyzing the coulomb diamonds in Fig. 2b, the capacitances of the quantum dot to the tip ($C_d$), backgate ($C_g$) and the surrounding conducting regions ($C_s$) were found to be in the ratio $C_g : C_s : C_d :: 1 : 37 : 36$ (see Supplementary Note 3). This implies that although $V_b$ is typically small compared to $V_g$, the tip is very efficient at gating the dot because of its larger capacitive coupling. This explains the positive slope of the oblique charging lines in the gate dependence map (Fig. 2a). The size of the quantum dot estimated by using a disc model ($61 \pm 4.8$ nm) is consistent with the size of the conducting island shown in Fig. 3a.

We also performed high resolution $dI/dV_b$ spatial mapping in the conducting region indicated by the dotted magenta square in Fig. 3a in order to directly image the confined carriers (Fig. 3e).

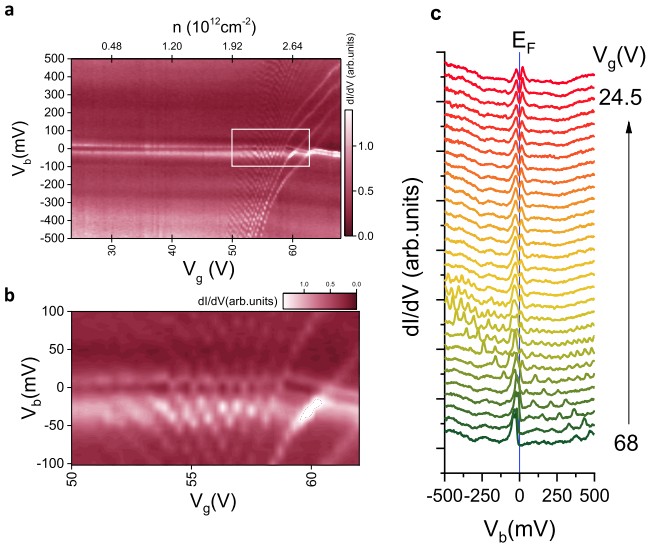

**Fig. 2 $dI/dV_b(V_b, V_g)$ maps at an AA region. a** Shows the $dI/dV_b(V_b, V_g)$ maps at an AA region. At high gate voltages, the flat bands appear as two bright lines below the $E_F$ at $V_b = 0$ mV, indicating that they are fully filled. As the gate voltage is reduced the $E_F$ intersects the flat bands and they begin to empty. Since the density of states in the flat bands is very high the bands, seen as two horizontal bright lines, look pinned to $E_F$ as they slowly empty. Note that the lowest applied gate voltage of 23.3 V is not enough to fully empty the bands. In addition to these expected spectroscopic features, a series of oblique lines can be seen in a narrow gate range of 50–68 V. When these lines intersect the $E_F$, a series of faint coulomb diamonds are seen. These additional spectroscopic features indicate the presence of a quantum dot in the system. **b** is a zoomed-in view of the region marked by the white rectangle in **a**. **c** Some selected $dI/dV_b$ spectra from the gate dependence map which show the confinement peaks in the spectra only when the Fermi level is near the band edge. The evolution of the individual confinement peaks with gate voltage which appears as the oblique lines in **a** can also be seen.

The three bright wavefronts in Fig. 3e correspond to three lowest energy states in the conducting region. See Supplementary Note 5 for more detailed analysis of the evolution of these wavefronts.

Next, we comment on the possible reasons behind the observed doping variation. The TBG rests on a thin hBN flake (23.5 nm). The hBN is exfoliated on top of a commercial highly p-doped Silicon chip with a thermally grown oxide layer (WaferPro, LLC). Silicon dioxide grown on Silicon wafers is known to suffer from dangling bonds, surface defects, and charged metallic impurities. In devices where graphene is directly supported by the silicon dioxide substrate, these imperfections give rise to charge puddles and to increased scattering[32–35]. The addition of chlorine gas or chlorine-based hydrocarbons during the growth of the silicon dioxide on top of silicon wafers has long been used in the manufacture of silicon-based metal-oxide-semiconductor field effect transistors as a way to passivate these metallic impurities[36,37]. The silicon wafer used in this device, however, was not passivated in this way.

The hBN, which is a wide gap dielectric, serves as a spacer which reduces the effect of the random impurity potential of the SiO₂ substrate[2,23,38]. Nonetheless, if the hBN is thin, like in our device (23.5 nm), some of the substrate-induced inhomogeneity can survive. Each of these charged impurities can act as a local gate and change the local charge density of the sample. Furthermore, hBN crystals themselves can host impurities and interstitial defects[39,40] which can dope the sample locally. Yet,

another possibility is trapped water or organic impurities which inevitably arise during sample fabrication.

Doping inhomogeneity caused by substrate disorder becomes especially important when the bulk Fermi level is tuned close to a band gap, such as the superlattice gaps at the edges of the flat bands or the gaps between Landau levels in a magnetic field. Once the flat bands are doped to integer fillings additional band gaps associated with correlation effects appear at the Fermi level. One should expect to also see quantum confinement near each of these fillings. However, in our STM measurements we did not see a clear signature of confinement when the Fermi level was tuned to integer fillings inside the flat band. There can be two reasons behind this observation: (1) as the Fermi level is tuned deeper inside the flat bands, the size of the quantum dot increases leading to decreased energy spacing of the confined states. This can we washed out by thermal excitations at the temperature of our experiment (4.9 K). Additionally, as the overall density of states is also greater inside the flat band, the disorder potential can be more effectively screened, leading to poorer confinement. (2) Another possibility is that the correlation gaps are not fully developed at the temperature of the experiment or that the gaps are smaller than our energy resolution. Additional experiments at much lower temperatures should be able to resolve these confinement effects.

To confirm our findings, we studied another TBG device, this time with a novel planar tunneling geometry (Fig. 4a) at 4.2 and 0.3 K. This device consists of an hBN encapsulated TBG with a local metallic backgate. The thick bottom hBN acts as a gate dielectric, while the top hBN is ultra-thin (<4 layers) and acts as a tunneling barrier. After locating a clean TBG region with atomic force microscopy (AFM), a metallic tunneling electrode is deposited on top of the thin hBN layer through a 600 nm diameter circular hole etched in a thick hBN (see "Methods"). A bias voltage is applied between the tunneling electrode and the TBG and the resulting tunneling current is amplified and measured. Similar to the STM measurement, we obtain the differential conductance ($dI/dV_b$) using a standard lock-in technique and tune the Fermi level of the sample by applying a gate voltage.

We estimate the twist angle from the measured energy separation of the VHS peaks[1]: $\Delta E_{VHS} \approx \left(\frac{2h\nu_F}{3a}\right)\theta - 2w_\perp$ where $h$ is Plank's constant, $\nu_F$ is the Fermi velocity of electrons in graphene, $a$ is the lattice constant of graphene, and $w_\perp \approx 110$ meV[3] is the interlayer tunneling parameter. A typical $dI/dV_b$ spectrum of TBG at $V_g = 0$ V is shown in Fig. 4b. Two peaks corresponding to the VHS which flank the charge neutrality point are consistent with STS results on TBG close to the magic angle. The measured $\Delta E_{VHS}$ of $48 \pm 2$ meV yields a twist angle of $(1.25 \pm 0.05)°$.

Assuming that the work function of TBG is close to the work function for Bernal stacked bilayer Graphene (4.7 V[41]) and that the work function of the tunneling electrode (Cr/Au) is close to that of Chromium ~4.5 eV[42], we can estimate a work function difference of the order of 100 meV between the TBG and the tunneling electrode. This work function difference results in the TBG becoming electron doped and creates a potential well, approximately the size of the tunneling electrode, where charge carriers can be confined. Similar results were seen previously in STM experiments on Bernal bilayer graphene[43], where a potential well was created by charging the impurities in hBN. This situation closely resembles the potential-wells observed in the STM device, albeit with different length scales and origins of the confinement potentials. The schematic diagram (Fig. 4d) illustrates this scenario. Indeed, gate dependence maps measured at 4.2 K (Fig. 4c) show Coulomb diamond-like features. On further cooling the sample to 0.3 K (Fig. 4e) and performing high resolution

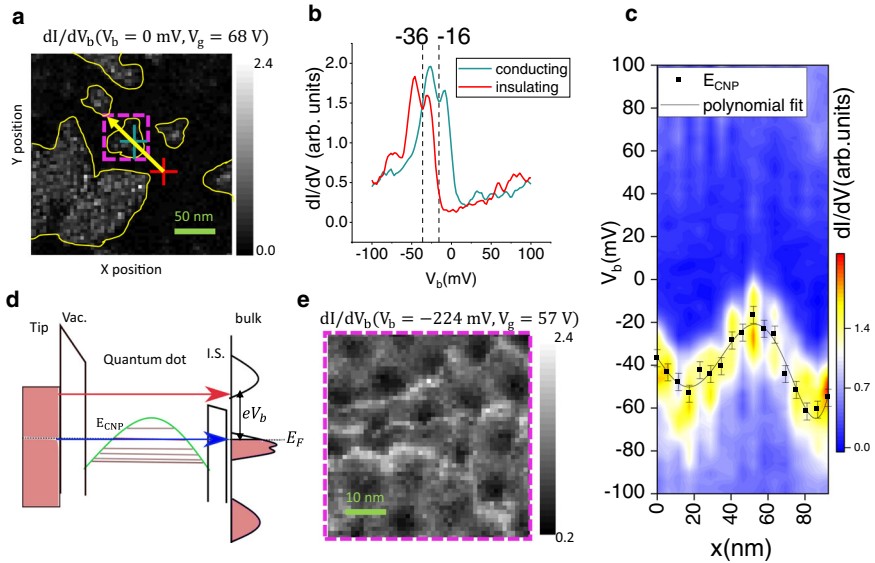

**Fig. 3 Evidence of local doping variation in the sample. a** Shows a $dI/dV_b(x, y, V_b = 0\ mV, V_g = 68\ V)$ spatial map of a $246 \times 246\ nm^2$ region (scale bar 50 nm). Several bright regions, indicating a high density of states, which are surrounded by dark regions, indicating a low density of states are visible. The thin yellow lines are a guide for the eyes. **b** Shows individual $dI/dV_b$ spectra measured at the positions marked by the red and green crosses in **a**. A lateral shift of the two spectra with respect to the $E_F$ indicates a difference in the local chemical potential. The approximate location of the charge neutrality point taken as the dip between the two VHS is also labeled. **c** is a line-cut across the conducting island in **a** in a direction indicated by the yellow arrow. The variation of the local charge neutrality point energy ($E_{CNP}$) as a function of position can be seen which indicates local doping variation. The black line is a 5th order polynomial fit. Error bars represent uncertainty in the determination of CNP due to finite energy resolution. The bright conducting regions surrounded by the darker insulating regions act like quantum dots. **d** A tunneling diagram of the system. There are two tunneling barriers: the large vacuum barrier between the tip and the dot and the shorter barrier between the dot and the bulk of the sample. The quantum dot is defined by a potential well which has the shape indicated by the local charge neutrality point. Confined states within this potential well are indicated by horizontal lines. There are low-energy electron tunneling events from the tip into the confined states of the quantum dot and then into the bulk of the sample (blue arrow). There also exist direct tunneling from the tip to the higher energy bands of the bulk of the sample (red arrow). **e** Shows a high resolution $dI/dV_b(x, y, V_b = -224\ mV, V_g = 57\ V)$ spatial map of the conducting island indicated by the magenta square in **a**. The three bright wavefronts are three single electron charging states confined in the bright island. The shape of the bright wavefronts approximately matches the shape of the bright island.

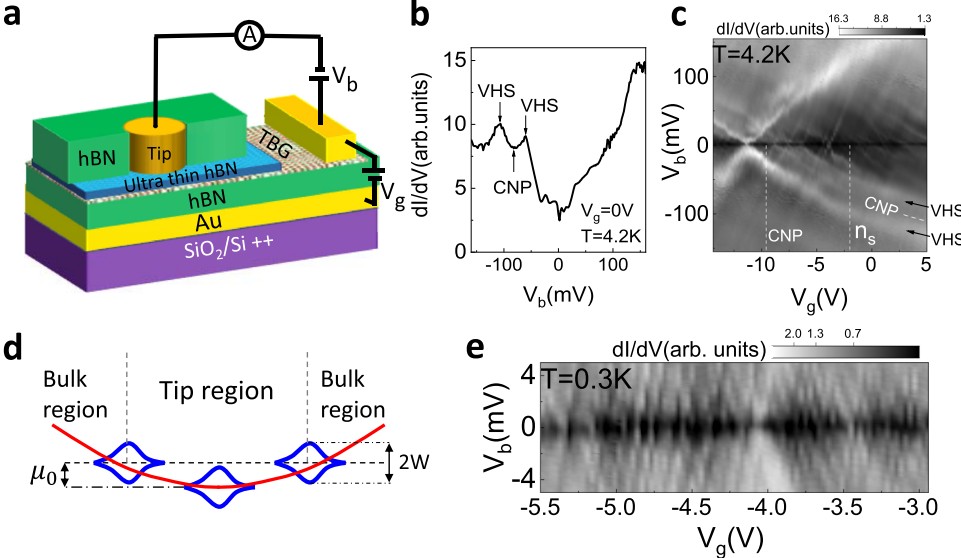

**Fig. 4 Carrier confinement in a TBG planar tunneling device. a** Schematic of the planar tunneling device. An ultra-thin hBN acts as a tunneling barrier between the metallic tip and the TBG. **b** Tunneling spectrum measured at 0 V backgate voltage showing the flat bands. The sample is highly n-doped because of the work function difference between the tunneling electrode and the TBG. **c** Gate dependence map of the STS. Flat bands can be seen as two bright parallel lines at negative bias. The other features are a result of confinement. **d** Schematic of the position dependence of the charge neutrality point. $\mu_0$ is the shift in the charge neutrality point under the tip compared to the bulk region. **e** High resolution $dI/dV_b$ gate dependence maps measured at 0.3 K showing coulomb diamonds, a result of electron confinement.

spectroscopic measurements, the resulting gate dependence shows well-developed Coulomb diamonds reflecting the carrier confinement in the device. The charging energy ($E_c$) extracted from the diamond-like features in Fig. 4e, ~2 meV, is much smaller than that observed in the STM device, consistent with a much larger dot size. The diameter of the QD in this case, estimated by using a 2D disc capacitance model[25] (~ 640 nm), agrees well with the size of the tip electrode. This supports the scenario of electron confinement due to the strong amplification of potential inhomogeneities in the flat band of magic-angle TBG.

## Discussion

We studied magic-angle TBG devices with STM/STS as well as in a planar tunneling geometry. Sharp peaks in the $dI/dV_b$ spectra near the full filling of the electron-side flat band and their evolution with gate voltage reveals carrier confinement in quantum dots generated by the substrate impurity potential or by the fabrication imposed potential well.

This work demonstrates that, owing to the unique ability of flat bands to amplify small doping inhomogeneities, magic-angle TBG samples are susceptible to small random disorder potentials, leading to a drastically modified charge density landscape. These effects, which are not directly observable in non-local measurements such as transport, have thus far been largely overlooked. However, as we have shown, they can nevertheless cause charging inhomogeneity revealing a hitherto hidden source of sample-to-sample variation that may contribute to the disparate findings in magic-angle TBG. Importantly, the flat band enabled confinement of charge carriers in graphene demonstrated here provides a design pathway for achieving on-off switching of graphene devices, which could not be otherwise attained.

## Methods

**STM device**. The TBG device was fabricated using the "twist and stack" method. polyvinyl alcohol (PVA) was spin coated on a silicon chip at 600 RPM followed by a 5 min bake on a hotplate at 90 °C. Poly (methyl methacrylate) (PMMA) A-11 (Kayaku Advanced Materials) was then spin coated at 3000 RPM on top of the PVA followed by a 30 min bake at 90 °C. Graphene was exfoliated using tape onto the PMMA/PVA coated chip. The PMMA/PVA stack was then peeled off from the silicon chip using a piece of tape with a small hole in it. The PVA backing layer was carefully removed with tweezers while leaving the PMMA membrane suspended across the hole in the tape window. If a suitable graphene flake was found on the PMMA membrane using an optical microscope, a small piece of poly-dimethylsiloxane (PDMS) 1 mm thick was place on the PMMA on the back side of the graphene. This assembly was then transferred to a glass slide creating a glass slide/PDMS/PMMA/graphene handle. An hBN flake was separately exfoliated on a highly p-doped silicon wafer capped by a 285 nm thick thermally grown (non-chlorinated) silicon oxide layer (Waferpro). The glass slide handle was attached to micromanipulators under an optical microscope and half of the graphene flake was brought into contact with the hBN flake. When the handle was lifted, the graphene in contact with the hBN was left on the hBN due to strong van der Waals interaction, while the other half remained on the PMMA membrane. The chip with the hBN was then rotated by the desired twist angle using a rotation stage and the other half of the graphene was peeled off onto the first half creating an exposed TBG supported by a thin flake of hBN.

Electrical contact was made to the graphene with metallic (Au/Ti) electrodes deposited using standard electron beam lithography techniques. A bias voltage was applied to the sample while the tip was grounded. The highly doped silicon was used as a backgate and enabled us to electrostatically tune the carrier density in the device. The measurements were conducted at a temperature of 4.9 K in a home built STM using chemically etched Tungsten tips which were tested on the gold electrode prior to scanning. Tunneling spectra were gathered in constant height mode using a lock-in detection technique by adding a 1–2 mV AC excitation to the DC bias at a frequency of 613 Hz. The sample was annealed in forming gas for 18 h at 270 °C before loading into the STM.

The LDOS maps are collected by a Nanonis software routine which measures the topography and $dI/dV_b$ spectrum in the same pass. In brief, at every position in the grid, the tip height is adjusted by the feedback loop based on an initial current and bias setpoint which is the same for all points in the grid. This height is recorded as the topography signal. Then, the feedback loop is disabled while keeping the tip height fixed and the point spectrum is measured by sweeping the sample bias. Thus, each spectrum is measured in a constant height mode but the height itself depends on the STM topography.

**Planar tunneling device**. The planar tunneling device consists of two parts, a top hBN layer containing etched holes plus an ultra-thin hBN tunneling layer(t-hBN), and a TBG/hBN stack with a local metal gate.

*Preparation of top hBN with etched holes*. Patterning of exfoliated hBN on SiO$_2$ was carried out using e-beam lithography. Then the patterned holes were etched out though CHF$_3$/O$_2$ plasma followed by a lift-off process.

*TBG/hBN stack with a local metal gate*. A PVA was spin coated on the silicon chip and baked at 80 °C for 10 min, followed by Polymethyl methacrylate (PMMA) spin-coating and baking at 80 °C for 30 min. Then graphene was exfoliated on the PMMA/PVA/silicon chip. The PMMA thin film was then peeled off from the PVA through the scotch tape with a precut window for subsequent transfer on a bottom hBN flake. The bottom hBN flake is transferred in advance onto a gold electrode as local gate and is annealed at 250 °C in Ar/H$_2$ for 6 h for surface cleaning. Next, part of the target graphene flake on PMMA film was brought in contact with the bottom hBN surface. Due to the stronger van der Waals interaction between graphene and hBN, the part of graphene flake in contact with the hBN could be detached from the PMMA upon lifting the film. Subsequently the substrate with G/hBN/gold stack was rotated by 1–1.2°. Then the rest part of the graphene flake on PMMA film was brought in contact with the G/hBN surface. The TBG/hBN/gold stack was annealed in forming gas (10% H$_2$, 90% Ar) at 235 °C for 6 h for better adhesion.

*Assembly of the hBN/t-hBN/TBG/gold stacks*. The hBN/t-hBN/TBG/hBN/gold stacks are assembled with the dry transfer method in a glovebox (Argon atmosphere), using a stamp consisting of polypropylene carbonate film and PDMS. The hBN flake with etched holes (30–50 nm thick) is firstly picked up. Then the t-hBN (<4 layers) is picked up by the hBN on the stamp. The hBN/t-hBN stack is then deposited onto the TBG/hBN/gold stack that is prepared separately in step (2). During the assembly of the stack the temperature is kept below 160 °C. AFM and electrostatic force microscopy are subsequently used to identify the region of the TBG[44] prior to depositing the electrical contacts (Cr/Au) for tunneling measurements. The tunneling measurements are carried out in He-3 system with a base temperature of 0.3 K. The $dI/dV_b$ measurements are using standard lock-in technique by adding AC excitation of 0.2–1 mV to the DC bias at 7.1 Hz.

**Growth of crystalline hBN**. Single-crystalline hBN was grown at high temperature and high pressure using a Rockland Research, cubic-multi-anvil press system. Elemental Mg and $^{11}$B, in a Mg:$^{11}$B 1:0.7 molar ratio, were placed in a 7 mm long, 5.7 mm inner diameter BN crucible with any empty space filled by extra BN powder. The crucible was then taken to 3.3 GPa at room temperature and heated to 1380 °C over 2 h. After dwelling at 1380 °C for 1 h, the system was cooled to 580 °C over 6 h and then quenched to room temperature. Once at room temperature the pressure was brought back to ambient conditions. Post growth, the crucible was removed from the pressure media and sealed in an amorphous silica ampoule. Excess Mg was distilled from the hBN by heating one end of the ampoule to 750 °C, while the other end hung out of the end of a horizontal tube furnace for 200 min. Thin, clear flakes of hBN with planar dimensions of up to $1 \times 2$ mm$^2$ would be separated from the growth crucible.

## Data availability

The data that support the findings of this study are available from the corresponding authors upon reasonable request.

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

## Acknowledgements

N.T. and X.L. acknowledge support from the U.S. DOE-BES grant DOE-FG02-99ER45742. S.W. and E.Y.A. acknowledge support from the Gordon and Betty Moore Foundation's EPiQS initiative grant GBMF9453. M.X. and P.C.C. were supported by the U.S. DOE-BES Division of MSE and the research was performed at the Ames Laboratory which is operated for the U.S. DOE by ISU under Contract No. DE-AC02-07CH11358. R.A.R. was supported by the Gordon and Betty Moore Foundation's EPiQS initiative grant GBMF4411.

## Author contributions

X.L. and N.T. fabricated and characterized the STM device. S.W. and Z.Z. fabricated and characterized the planar tunneling device. M.X., R.A.R., and P.C.C. grew the hBN crystal used to make the devices. N.T. and E.Y.A. wrote the paper with inputs from all authors. E.Y.A. supervised the research.

## Competing interests

The authors declare no competing interests.
