## [Peer Review File · Nature Communications]

REVIEWER COMMENTS

Reviewer #1 (Remarks to the Author):

Tilak et. al. have used STM and scanning tunneling spectroscopy measurements for experimentally demonstrating the effect of charge disorder when magic angle graphene is doped near the flat band edge. The spatial maps provide a convincing illustration of the formation of metallic and insulating regions and the conclusion is well supported by the observation of Coulomb diamonds and formation of quantum dots. The authors suggest the use of random charge disorder as an alternative means of electronic confinement.

The physics of moiré heterostructures and twisted graphene is an active area of research and this work is likely to be of interest to the condensed matter community.

1. While the observation of this effect is convincingly argued, the existence of local doping variations leading to confinement is not unexpected. As the authors have pointed out, this will occur whenever there is a band gap, such as in Landau levels. Its observation near a single particle gap is not specific to graphene twisted at ~ 1 degree, but should be seen at other angles as well where single particle gaps exist without flat bands. Is it possible to see a signature of this effect inside the flat band at the correlated gaps? Can the authors comment on how this confinement is specific to flat bands?

2. Can the authors estimate the random charge disorder in their samples and how it compares to values reported elsewhere, such as refs 24 and 21? If the charge disorder is quantified, the planar tunnel junction measurement might also be useful to demonstrate that the effect is not limited to unencapsulated samples used for STM measurements, but is also seen in fully encapsulated graphene that might presumably have lower disorder.

3. What was the thickness of the bottom h-BN in the STM experiment? It has been described as “very thin” in the manuscript on line 163.

4. The manuscript has a number of typos which should be addressed: (a) the caption of fig 1d mentions a dotted line which not present (b) supplementary note 5 mentions “Main Figure 2c “ which does not exist (c) line 139, figure S3 should be S5, (d) figure 3c – label x and y axis (e) figure S4j

– axis label not present, fonts are too small (f) add colorscale in figure 3b (g) references 21 and 23 are duplicates

Reviewer #2 (Remarks to the Author):

In this work, the authors performed STM and planar tunneling experiments in magic-angle twisted bilayer graphene, and found that the edge of the flat bands are highly sensitive to disorder from the substrate due to the small bandwidth. They further demonstrated that this disorder leads to confinement of carriers, which manifest as Coulomb diamonds in the tunneling spectra.

Overall, the paper is well-presented and easy to understand, but I find the data quality in the paper to be below average, and the results from this paper adds little new knowledge to the field. For these reasons, I cannot recommend publication of this work to Nature Communications. I wish the authors to consider the following suggestions/address the following comments if they wish to resubmit to another journal, in a revised form.

1. The tunneling spectra appears to be of quite low quality comparing to other STM works in magic-angle TBG that have been published so far. For example, the authors claims that 'There are gap-like features of ~ 20 - 30 meV between the flat bands and the remote bands on either side.' in Fig. 1c, but to my eyes only the vHs peaks are really visible. The authors should specify what is the temperature of these measurements.

2. There seem to not have much gate voltage dependence judging from the lower voltage part of Fig. 2a, again comparing with previous STM works, including one of their own (Nature 573, 91–95(2019)). Is there any particular explanation for this? Could it be the sample is not well-gated?

3. One of the main claim of the authors is that charge inhomogeneity due to the substrate potential can obscure magic-angle physics, which is not accounted for in transport experiments. They showed in Fig. 3 that there are puddles of compressible and incompressible states. However, the tunneling spectra shown in Fig. 3b indicate that the disorder potential is almost 50 meV, which is in fact larger than then bandwidth of MATBG itself! It is unlikely that in any state-of-art MATBG devices the disorder will be this large. In fact, a recent transport experiment (arXiv:2008.12296) demonstrated energy resolution of ~ 1 meV even though averaging over a sample size of a few square micrometers.

It appears to me that the charge disorder discovered by the authors are specific to their samples and cannot be generalized at all to the general case.

4. The authors spends almost an entire page explaining how Coulomb blockade and Coulomb diamond works, which can be found in textbooks and adds little information.

5. Parts of the text are somewhat repetitive, for example on line 98-99 the authors talked about charge confinement in graphene needs a high magnetic field, and this is mentioned again on line 170-171.

6. In Fig. 4c, there are a lot of features in parallel. It's not clear to me how the authors identify what is vHs and what is CNP. Furthermore, the authors used several methods to try to determine the twist angle (which again I think shall be left for SI). The second method relies on gating towards the full-filling gap at $\pm n$ s, which I cannot see any feature of such in Fig. 4c. The authors mentioned "half filled at $V_g = -9.7V$ ", but how do they know it's half-filled? The only visible features are the vHs, which do not necessarily lie at half-filling. While it is not a big issue about the main claim of this paper, this clearly shows that the data analysis performed in this paper is far from rigorous.

Reviewer #3 (Remarks to the Author):

Tilak et al. present low temperature scanning tunneling spectroscopy (STS) and planar tunneling spectroscopy (PTS) measurements of a magic angle twisted bilayer graphene (TBG) sample that is

supported by a hexagonal boron nitride (hBN) flake. Both measurements utilized samples that are equipped with gating capabilities. For the STS study, tunneling spectroscopy was acquired as a function of gate and sample bias voltages. Clear signatures of the magic angle TBG electronic structure were seen, which has been previously reported. The new observed features, which are the focus of this manuscript, are diagonal lines and faint diamonds in a $dI/dV_b(V_b, V_g)$ plot. To further understand these features, the authors performed constant energy dI/dV_b maps with sample and gate voltage configurations consistent with the diagonal lines in their $dI/dV_b(V_b, V_g)$ plot. These data reveal conductive (compressible) islands that are surrounded by insulating (incompressible) regions. Such phenomena give rise to charge carrier confinement. From the dI/dV_b maps, the compressible regions manifest at length scales larger than the moiré wavelength. The authors attribute the emergence of these conductive islands within insulating regions to an inhomogeneous potential landscape created by the supporting substrate. The authors propose that such a landscape could result from charged defects in the thin underlying hBN, impurities in the supporting SiO₂, or charge traps at the TBG/hBN interface. The authors noted that the inhomogeneous potential landscape is most influential when the TBG flat bands are nearly filled. To confirm this interpretation the authors performed PTS measurements of a TBG sample. In these measurements the tunnel probe imparted a localized potential in the TBG sample, hence mimicking the inhomogeneous potential in the STM experiment. The spectral characterization from PTS revealed similar features as the STM experiments.

The work presented by Tilak et al. is very timely because magic angle TBG is an exceedingly popular platform for experimentalists and theoreticians in condensed matter physics. In addition, the authors' work addresses charge inhomogeneity, a topic that has been generally overlooked. Yet, charge inhomogeneity is important because it might be a contributing factor to the reported variability between different samples and different groups. Because of these points I am inclined to recommend publication of this manuscript in Nature Communications if the authors are able to address my specific questions/concerns below.

(1) The authors should perform further characterization of the confined carriers. For example, the authors should perform $dI/dV_b(V_s, x)$ line scans of the conducting islands shown in Figure 3a similar to the work by F. Ghahari et al Science 2017. This would enable direct measurement of the level spacing. Additionally, this measurement will provide a deeper understating of the confinement potential profile.

(2) The authors should discuss the reproducibility of their PTS results. Previous work by S. Jung et al Nano Letters 2017 and J. Davenport et al. APL 2019 have shown that PTS devices are comparable to STS results but can display large variability between samples. Did the authors measure other PTS devices with different tunneling probe dimensions? Additional PTS measurements on different devices would strengthen the connection made by the authors between the STM experiments.

(3) The authors should provide more details for their dI/dV_s maps in which insulating regions were scanned. Did the authors use a constant height mode for these scans?

(4) The authors should provide better labeling and additional information within their figures. This will enhance the accessibility of the article for a broader audience and improve communication of the experimental findings. The following should be included: (i) a stacking diagram in Figure 1 similar to Figure 1 in the work by Kerelsky et al. Nature 2019; (ii) labeling of electronic structure features in the spectra in Figure 1c; (iii) a magic angle TBG band diagram associated with their sample in Figure 1.

(5) Key references from previous works pertaining to the interpretation of the manuscript are missing. This includes S. Jung et al. Nature Physics 2011, which studied disorder induced QDs at high magnetic fields in graphene/SiO₂ devices. A reference to D. Wong et al Nature Nano. 2015 should also be included because this work imaged charged defects in hBN with STM. These defects are likely contributing to the charge inhomogeneity that the authors see in their STM experiments.

Reviewer #1 (Remarks to the Author):

Tilak et. al. have used STM and scanning tunneling spectroscopy measurements for experimentally demonstrating the effect of charge disorder when magic angle graphene is doped near the flat band edge. The spatial maps provide a convincing illustration of the formation of metallic and insulating regions and the conclusion is well supported by the observation of Coulomb diamonds and formation of quantum dots. The authors suggest the use of random charge disorder as an alternative means of electronic confinement.

The physics of moiré heterostructures and twisted graphene is an active area of research and this work is likely to be of interest to the condensed matter community.

1. While the observation of this effect is convincingly argued, the existence of local doping variations leading to confinement is not unexpected. As the authors have pointed out, this will occur whenever there is a band gap, such as in Landau levels. Its observation near a single particle gap is not specific to graphene twisted at ~ 1 degree, but should be seen at other angles as well where single particle gaps exist without flat bands. Is it possible to see a signature of this effect inside the flat band at the correlated gaps? Can the authors comment on how this confinement is specific to flat bands?

As the reviewer correctly pointed out, single particle gaps at the flat band edges in small angle TBG are an essential ingredient in order to observe clear quantum confinement effects. These single-particle band gaps, which arise due to lattice relaxation effects, have been experimentally observed for TBG when the twist angle is less than about 3-4 degrees. So, we do expect to see signatures of confinement at other small twist angles which are larger than the magic angle but smaller than 3-4 degrees. The sample that we studied in this work was prepared such that the twist angle is very close to the magic-angle so we do not have data for larger twist angles. This could be explored in future experiments.

Increasing the flatness of the isolated bands enhances these confinement effects. If the isolated bands are wider, like at larger twist angles, then the DOS near the band edges drops off more gently with energy. Since there are very few states close to the band edge, the Fermi level can be tuned deep within such a wide band with very little gate voltage. This causes the size of the Quantum dot to increase very quickly, resulting in a reduction in the energy spacing of the confined states which can even get washed out by thermal excitations. That makes it harder to resolve these confinement effects for wider bands. For flatter bands the size of the Quantum dots is less sensitive to the Fermi level. This is illustrated in the simplified sketch below.

In our STM measurements we did not see a pronounced signature of confinement when the Fermi level was tuned to integer moiré fillings where correlation effects open a gap inside the flat bands. There can be two reasons behind this observation: 1. As the Fermi level is tuned deeper inside the flat bands, the size of the quantum dot increases leading to decreased energy spacing of the confined states. This can be washed out by thermal excitations at the temperature of our experiment. Additionally, as the overall density of states is also greater inside the flat band the disorder potential can be more effectively screened leading to poorer confinement. 2. We know that the correlation gaps, which are of order 2meV^{-1} aren't fully developed at the temperature of the experiment (4.9 K) these factors could be overcome by further lowering the measurement temperature. In fact, in a recently published STM work (Nuckolls et al, Nature 2020) on MATBG, which was carried out at a much lower temperature of 200 mK in a dilution refrigerator, carrier confinement effects were observed in the Chern insulating gap near +1 filling state under a 6T magnetic field. The authors attributed their observation to tip induced charge disorder. In our planar tunneling junction (PTJ) measurements, we also used a tip induced charge island in order to replicate confinement effects seen in our STM device and used a much lower temperature of 300 mK. Unlike the STM data, in the PTS data we saw that the coulomb diamond features exist in the entire accessible range of gate voltage, when the fermi level is tuned inside the flat bands.

2. Can the authors estimate the random charge disorder in their samples and how it compares to values reported elsewhere, such as refs 24 and 21? If the charge disorder is quantified, the planar tunnel junction measurement might also be useful to demonstrate that the effect is not limited to unencapsulated samples used for STM measurements but is also seen in fully encapsulated graphene that might presumably have lower disorder.

We estimate that the standard deviation of the charge disorder in the unencapsulated STM sample is about $3 \times 10^{10} e/cm^2$. The details of the analysis have been included in the supplementary information. This is comparable to the charge disorder value reported in ref 21 and 24 for MATBG devices.

3. What was the thickness of the bottom h-BN in the STM experiment? It has been described as “very thin” in the manuscript on line 163.

The thickness of the bottom hBN is 23.5 nm, measured using AFM topography as shown below. We have also added this information to the manuscript.

4. The manuscript has a number of typos which should be addressed: (a) the caption of fig 1d mentions a dotted line which not present (b) supplementary note 5 mentions “Main Figure 2c “ which does not exist (c) line 139, figure S3 should be S5, (d) figure 3c – label x and y axis (e) figure S4j – x-axis label not present, fonts are too small (f) add colorscale in figure 3b (g) references 21 and 23 are duplicates

We thank the referee for bringing these typos to our attention. We have now made the following corrections-

- (a) A dashed vertical line has been added to Fig 1d.
- (b) This typo has been fixed now.
- (c) This typo has been fixed now.
- (d) Added a X and Y label to the axis.
- (e) Increased font size for easier visibility
- (f) Added a legend to Fig 3b.
- (g) Removed duplicate references.

Reviewer #2 (Remarks to the Author):

In this work, the authors performed STM and planar tunneling experiments in magic-angle twisted bilayer graphene and found that the edge of the flat bands are highly sensitive to disorder from the substrate due to the small bandwidth. They further demonstrated that this disorder leads to confinement of carriers, which manifest as Coulomb diamonds in the tunneling spectra.

Overall, the paper is well-presented and easy to understand, but I find the data quality in the paper to be below average, and the results from this paper adds little new knowledge to the field. For these reasons, I cannot recommend publication of this work to Nature Communications. I wish the authors to consider the following suggestions/address the following comments if they wish to resubmit to another journal, in a revised form.

1. The tunneling spectra appears to be of quite low quality comparing to other STM works in magic-angle TBG that have been published so far. For example, the authors claims that 'There are gap-like features of ~20-30 meV between the flat bands and the remote bands on either side.' in Fig. 1c, but to my eyes only the vHs peaks are really visible. The authors should specify what is the temperature of these measurements.

It is indeed difficult to see the single particle gaps surrounding the flat bands in the spectrum in Fig 1c. We do see these gap-like features more clearly in other spectra which we have now included in the SI.

All STM/STS data was gathered at a sample temperature of 4.9 K as noted in the methods section. We have now also added this information to figure captions for clarity. Since the correlation gaps at integer fillings of the flat band are at most $\sim 2\text{meV}$ it is therefore not surprising that the gaps should be smeared out at 4.9K. In other words this is not a reflection of ‘low data quality’ but rather of the inconvenient and unavoidable physics of thermal excitations.

2. There seem to not have much gate voltage dependence judging from the lower voltage part of Fig. 2a, again comparing with previous STM works, including one of their own (Nature 573, 91–95(2019)). Is there any particular explanation for this? Could it be the sample is not well-gated?

The gate voltage dependence looks weak in the lower voltage part of Fig 2a because the Fermi level is still inside the flat band in this voltage range. As discussed in the main text, it takes about 80 V gate voltage change to tune the band filling from $\nu = +4$ to $\nu = -4$. This range is strictly dictated by electrostatics and will depend directly on the thickness of the insulating substrate as detailed in the text. In Fig 2a the Flat band is at a filling of +4 near a gate voltage of 60 V. This implies that it will be completely empty around a gate voltage of $60-80 = -20\text{V}$. We collected data in a narrow gate range from 68 to 23.3 V in order to increase the resolution in the gate voltage and see the charging lines more clearly while keeping the acquisition time reasonably low.

As an additional proof that the gate is indeed well-connected we have added in the SI, another dI/dV gate map taken at an AA site near the position where the data in Fig 1 was measured. The Fermi level at $V_b=0$ mV is clearly tuned from one side of the band to the other.

Furthermore, we rely on a capacitance-based technique² to navigate the STM tip to our O (μm) size samples on a O(cm) size insulating wafer. This technique involves applying an AC voltage to the sample and simultaneously applying the same AC voltage out of phase by 180 degrees to the Silicon back gate. Since we used this technique successfully to navigate to the sample, we have no doubt that the back gate was well connected during the experiment.

3. One of the main claim of the authors is that charge inhomogeneity due to the substrate potential can obscure magic-angle physics, which is not accounted for in transport experiments. They showed in Fig. 3 that there are puddles of compressible and incompressible states. However, the tunneling spectra shown in Fig. 3b indicate that the disorder potential is almost 50 meV, which is in fact larger than the bandwidth of MATBG itself! It is unlikely that in any state-of-art MATBG devices the disorder will be this large. In fact, a recent transport experiment (arXiv:2008.12296) demonstrated energy resolution of ~ 1 meV even though averaging over a sample size of a few square micrometers. It appears to me that the charge disorder discovered by the authors are specific to their samples and cannot be generalized at all to the general case.

Analyzing the LDOS map in Fig 3a shows that the standard deviation of the E_{CNP} is ~ 15 meV in this region. The analysis also revealed that the curves in Fig 3b which display a E_{CNP} shift of ~ 50 meV are outliers and we have replaced the curves with spectra taken at nearby points which better represent the distribution of the CNP.

We estimate that the standard deviation of the charge disorder in the unencapsulated STM sample is about $3 \times 10^{10} e/\text{cm}^2$. The details of the analysis have been included in the supplementary information. This is comparable to the charge disorder value reported in ref 21 and 24 for MATBG devices. As noted in the manuscript, this disorder can arise from various sources during sample fabrication or from impurities in the Silicon and hBN substrate.

We agree with the referee that state-of-the-art transport devices, including those measured in our lab, have observed lower charge disorder, possibly due to complete encapsulation in hBN. In global transport measurements, scattering from disorder manifests itself as a broadening of the conductance features. Our work complements transport studies by taking a closer look at the effects of disorder on a local scale. We see that the disorder can form patches of conducting or insulating regions on the 10-100 nm scale while the rest of the sample shows “normal” behavior. As

long as there are large enough conducting paths between macroscopic transport electrodes, correlation effects in MATBG devices will still be visible on a global scale. On the local scale, however, as we show in this work, they are easily disrupted by doping disorder. Further experiments where transport and STM measurements are conducted on the same sample are needed to shed more light on this effect.

Our additional analysis of the charge disorder highlights that charge disorder, whatever be the underlying cause, presents unique challenges. In fact, in a newly published STM experiment (Nuckolls et al, Nature 2020) on MATBG, which was carried out at 200 mK, similar carrier confinement effects were also observed in the Chern insulating gap near +1 filling state under a 6T magnetic field. The authors attributed their observation to tip induced charge disorder.

4. The authors spends almost an entire page explaining how Coulomb blockade and Coulomb diamond works, which can be found in textbooks and adds little information.

We have significantly shortened this part and moved the details to the SI.

5. Parts of the text are somewhat repetitive, for example on line 98-99 the authors talked about charge confinement in graphene needs a high magnetic field, and this is mentioned again on line 170-171.

We thank the referee for their observation. We have modified the text to avoid repetition.

6. In Fig. 4c, there are a lot of features in parallel. It's not clear to me how the authors identify what is vHs and what is CNP. Furthermore, the authors used several methods to try to determine the twist angle (which again I think shall be left for SI). The second method relies on gating towards the full-filling gap at $\pm n$ s, which I cannot see any feature of such in Fig. 4c. The authors mentioned "half filled at $V_g = -9.7V$ ", but how do they know it's half-filled? The only visible features are the vHs, which do not necessarily lie at half-filling. While it is not a big issue about the main claim of this paper, this clearly shows that the data analysis performed in this paper is far from rigorous.

Thanks for the referee's suggestions. The discussion of determination of the twist angle has been moved to the supplementary information. As shown in Fig. 4b, the dI/dV as a function of bias voltage (V_b) at fixed gate voltage (V_g) is consistent with the typical STS result of twisted bilayer graphene. The VHS and CNP have been marked for clarity. By applying various gate voltages, the Fermi level could be tuned. A mapping of $dI/dV(V_b, V_g)$ is obtained as shown in Fig. 4c. The trajectory of VHS and CNP with V_b and V_g can be traced. We have added additional labels in Fig. 4c which guide the eyes towards the relevant spectroscopic features. When the Fermi level ($V_b=0V$) is tuned to lie outside the VHS, i.e. $V_g = -(2\pm 1)V$, the band is fully-filled. When the Fermi level ($V_b=0V$) is tuned to lie at the CNP, i.e. $V_g = -9.7V$ here, the bands are half-filled, i.e., 4 e- per moiré site.

Reviewer 3:

Tilak et al. present low temperature scanning tunneling spectroscopy (STS) and planar tunneling spectroscopy (PTS) measurements of a magic angle twisted bilayer graphene (TBG) sample that is supported by a hexagonal boron nitride (hBN) flake. Both measurements utilized samples that are equipped with gating capabilities. For the STS study, tunneling spectroscopy was acquired as a function of gate and sample bias voltages. Clear signatures of the magic angle TBG electronic structure were seen, which has been previously reported. The new observed features, which are the focus of this manuscript, are diagonal lines and faint diamonds in a $dI/dV_b(V_b, V_g)$ plot. To further understand these features, the authors performed constant energy dI/dV_b maps with sample and gate voltage configurations consistent with the diagonal lines in their $dI/dV_b(V_b, V_g)$ plot. These data reveal conductive (compressible) islands that are surrounded by insulating (incompressible) regions. Such phenomena give rise to charge carrier confinement. From the dI/dV_b maps, the compressible regions manifest at length scales larger than the moiré wavelength. The authors attribute the emergence of these conductive islands within insulating regions to an inhomogeneous potential landscape created by the supporting substrate. The authors propose that such a landscape could result from charged defects in the thin underlying hBN, impurities in the supporting SiO₂, or charge traps at the TBG/hBN interface. The authors noted that the inhomogeneous potential landscape is most influential when the TBG flat bands are nearly filled. To confirm this interpretation the authors

performed PTS measurements of a TBG sample. In these measurements the tunnel probe imparted a localized potential in the TBG sample, hence mimicking the inhomogeneous potential in the STM experiment. The spectral characterization from PTS revealed similar features as the STM experiments. The work presented by Tilak et al. is very timely because magic angle TBG is an exceedingly popular platform for experimentalists and theoreticians in condensed matter physics. In addition, the authors' work addresses charge inhomogeneity, a topic that has been generally overlooked. Yet, charge inhomogeneity is important because it might be a contributing factor to the reported variability between different samples and different groups. Because of these points I am inclined to recommend publication of this manuscript in Nature Communications if the authors are able to address my specific questions/concerns below.

- (1) The authors should perform further characterization of the confined carriers. For example, the authors should perform $dI/dV_b(V_b, x)$ line scans of the conducting islands shown in Figure 3a similar to the work by F. Ghahari et al Science 2017. This would enable direct measurement of the level spacing. Additionally, this measurement will provide a deeper understating of the confinement potential profile.

We thank the referee for their suggestion. We have now added a plot to Fig 3 which shows the potential profile across the conducting island shown in Fig 3a. The overall shape of the potential is clearly visible. Unfortunately, the spatial resolution of this large area map is too small to reveal individual charging lines within the conducting island.

We have a much higher resolution dI/dV map of this region which was measured at a lower gate voltage of 57 V where the charging lines are visible. We have included a line profile across this map in the supplementary information. The level spacing can be seen in these maps as suggested by the referee.

- (2) The authors should discuss the reproducibility of their PTS results. Previous work by S. Jung et al Nano Letters 2017 and J. Davenport et al. APL 2019 have shown that PTS devices are comparable to STS results but can display large variability between samples. Did the authors measure other PTS devices with different tunneling probe dimensions? Additional PTS measurements on different devices would strengthen the connection made by the authors between the STM experiments.

Thanks for the referee's suggestion. We measured another PTS device (#2) with the same tunneling probe dimension at 77K. We found that a heavy doping effect occurs in this device similar to that in the device (#1) reported in the manuscript. The STS results at various gate voltages at 77K of both devices are shown below. This figure has been added to the supplementary information.

Figure: $dI/dV(V_b, V_g)$ of device #1, #2 at 77K

(3) The authors should provide more details for their $dI/dV_b(V_b, V_g)$, maps in which insulating regions were scanned. Did the authors use a constant height mode for these scans?

The LDOS maps are collected by a Nanonis software routine which measures the topography and dI/dV_b spectrum in the same pass. In brief, at every position in the grid, the tip height is adjusted by the feedback loop based on an initial current and bias setpoint which is the same for all points in the grid. This height is recorded as the topography signal. Then, the feedback loop is disabled while keeping the tip height fixed and the point spectrum is measured by sweeping the sample bias. Thus, each spectrum is measured in a constant-height mode but the height itself depends on the STM topography. We have added this information to the methods section for clarity.

(4) The authors should provide better labeling and additional information within their figures. This will enhance the accessibility of the article for a broader audience and improve communication of the experimental findings. The following should be included: (i) a stacking diagram in Figure 1 similar to Figure 1 in the work by Kerelsky et al. Nature 2019; (ii) labeling of electronic structure features in the spectra in Figure 1c; (iii) a magic angle TBG band diagram associated with their sample in Figure 1.

We appreciate this suggestion and have made several changes to all the figures to make them easier to read. We have also added the requested panels and labels in Fig 1 to make it more informative.

(5) Key references from previous works pertaining to the interpretation of the manuscript are missing. This includes S. Jung et al. Nature Physics 2011, which studied disorder induced QDs at high magnetic fields in graphene/SiO₂ devices. A reference to D. Wong et al Nature Nano. 2015 should also be included because this work imaged charged defects in hBN with STM. These defects are likely contributing to the charge inhomogeneity that the authors see in their STM experiments.

Thank you for pointing us to these works. These references were omitted in error. We have now added them in.

References

- 1 Wu, S., Zhang, Z., Watanabe, K., Taniguchi, T. & Andrei, E. Y. Chern Insulators and Topological Flat-bands in Magic-angle Twisted Bilayer Graphene. *arXiv e-prints*, arXiv:2007.03735 (2020).
- 2 Li, G., Luican, A. & Andrei, E. Y. Self-navigation of a scanning tunneling microscope tip toward a micron-sized graphene sample. *Review of Scientific Instruments* **82**, 073701, doi:10.1063/1.3605664 (2011).

REVIEWERS' COMMENTS

Reviewer #1 (Remarks to the Author):

The response and the revised version of the manuscript appropriately address the questions raised by the referees. I recommend the paper for publication in Nature Communications.

Reviewer #2 (Remarks to the Author):

The authors have satisfactorily answered my previous questions, and as such the paper becomes more readable overall. I still have some concerns regarding the novelty of the work, but as the other reviewers seem ok I'll leave the decision to the editor to decide.

Reviewer #3 (Remarks to the Author):

The authors have addressed all of my concerns and criticisms. I recommend publication of this manuscript.